# MutualNeRF: Improve the Performance of NeRF under Limited Samples with Mutual Information Theory

## Abstract

This paper introduces MutualNeRF, a framework enhancing Neural Radiance Field (NeRF) performance under limited samples using Mutual Information Theory. While NeRF excels in 3D scene synthesis, challenges arise with limited data and existing methods that aim to introduce prior knowledge lack theoretical support in a unified framework. We introduce a simple but theoretically robust concept, Mutual Information, as a metric to uniformly measure the correlation between images, considering both macro (semantic) and micro (pixel) levels. For sparse view sampling, we strategically select additional viewpoints containing more non-overlapping scene information by minimizing mutual information without knowing the ground truth images beforehand. Our framework employs a greedy algorithm, offering a near-optimal solution for this task. For few-shot view synthesis, we maximize the mutual information between inferred images and ground truth, expecting inferred images to gain more relevant information from known images. This is achieved by incorporating efficient, plug-and-play regularization terms. Experiments under limited samples show consistent improvement over state-of-the-art baselines in different settings, affirming the efficacy of our framework.

## 1 Introduction

NeRF (Mildenhall et al., 2020) (Neural Radiance Fields) is an advanced technique in computer graphics and computer vision that enables highly detailed and photorealistic 3D reconstructions of scenes from 2D images (Zhang et al., 2020; Park et al., 2021; Pumarola et al., 2021). It represents a scene as a 3D volume, where each point in the volume corresponds to a 3D location and is associated with a color and opacity. The key idea behind NeRF is to learn a deep neural network that can implicitly represent this volumetric function, allowing the synthesis of novel views of the scene from arbitrary viewpoints.

Although NeRF can synthesize high-quality images, it often relies on a large amount of high-quality training data (Yu et al., 2021b). The performance of NeRF drastically decreases when the number of training data is reduced. To mitigate this, existing strategies include adding new samples to the dataset and integrating regularization terms to introduce prior knowledge. For adding new samples, ActiveNeRF (Pan et al., 2022) aims to supplement the existing training set with newly captured samples based on an active learning scheme. It incorporates uncertainty estimation into a NeRF model and selects the samples that bring the most information gain. However, its reliance on the variance shift between prior and posterior distributions as a metric for information gain is somewhat speculative and can lead to unreliable outcomes. Regarding regularization, a plethora of studies (Niemeyer et al., 2022; Yang et al., 2023; Yu et al., 2021b; Jain et al., 2021) have explored the integration of prior or domain-specific knowledge to facilitate high-quality novel view synthesis and enhance generalization capabilities, even with limited training data. However, many of these methods lack theoretical support, hindering their explanation and optimization within a unified framework.

Confronting challenges in the few-shot scenarios, we introduce a theoretically robust and computationally efficient strategy addressing two pivotal tasks: sparse view sampling and few-shot view synthesis. Sparse view sampling targets acquiring training images from a selection of candidate views without knowing their ground truth images. Our strategy intuitively emphasizes minimizing

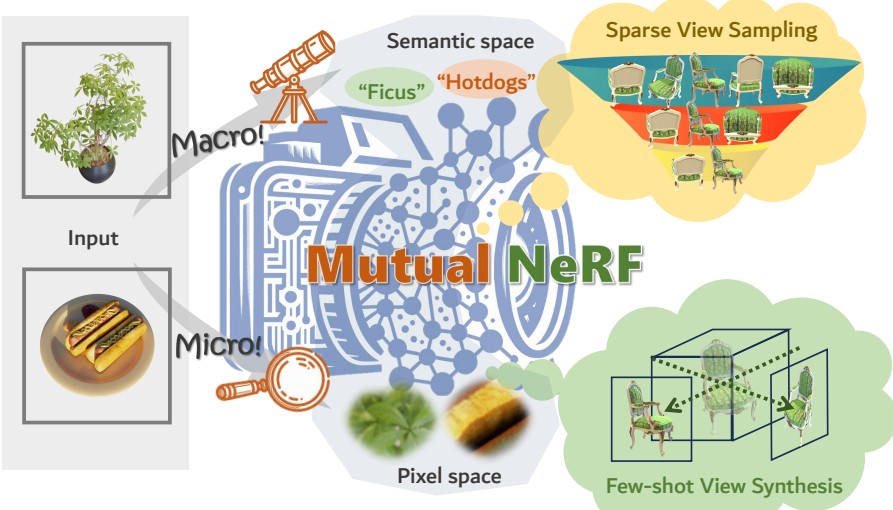

**Figure 1: The overview of MutualNeRF.** We introduce a novel and generic NeRF framework, comprehensively integrating mutual information from macro (semantic space) and micro perspectives (pixel space). This dual-perspective framework adeptly addresses challenges in sparse view sampling and few-shot view synthesis.

the correlation between training images for more unique information. Transitioning to few-shot view synthesis, our strategy involves training a NeRF model on a predetermined training set, aiming to maximize the correlation between inferred images and ground truth in the view synthesis process.

In this work, we introduce the concept of Mutual Information as an interpretable metric to model correlation. This concept is inspired by TupleInfoNCE (Liu et al., 2021), which effectively models mutual information across different modalities to enhance multi-modal fusion. Mutual information serves as a metric for quantifying the uncertainty between variables, especially pertinent in the NeRF context. On the one hand, it can guide us in selecting inputs to encapsulate maximal information with fewer images. On the other hand, it assesses the uncertainty of unknown view synthesis given known views.

We approach mutual information from a two-fold perspective: the macro perspective and the micro perspective. The macro perspective focuses on the correlation in semantic features, particularly employing the CLIP (Radford et al., 2021) method for semantic space distance, while the micro perspective in pixel space deals with the decomposition of relative information between images based on ray differences. To ensure feasibility and computational efficiency, pixel space distance is correlated with the Euclidean distance between camera positions and RGB color differences. Furthermore, we take into account multiple training set images for unknown scenes, introducing mutual information for multiple images.

Leveraging mutual information as the metric, our novel algorithmic framework can tackle board challenges in sparse view sampling by introducing new samples with less mutual information, and few-shot view synthesis by adding new regularization terms to increase the mutual information between the inferred images and the ground truth.

In sparse view sampling, the task is to select a subset of images from a candidate view set with unknown ground truth to supplement the training process. Ground truth is revealed only after selection, following the active learning framework. Our strategy focuses on minimizing redundancy in the selected views to maximize information gain. We introduce a computationally efficient greedy algorithm with a look-ahead strategy, which functions as a near-optimal solution. This algorithm iteratively selects images based on their contribution to unexplored information. The selection criteria combine semantic space distances, as derived from CLIP, with pixel space distance, calculated using the Euclidean distance between camera positions.

For few-shot view synthesis, the task is to directly train a NeRF model with limited and fixed training samples. we aim to develop efficient, plug-and-play regularization terms for the training procedure. The objective is to maximize the mutual information between the training images and randomly

rendered images, expecting inferred images to gain more relevant information from known images. We assess semantic space distance by CLIP as the macro regularization term. As camera position is invariant to the parameter of the NeRF, we utilize a computationally efficient metric dependent on both camera positioning and network parameters. It serves as the micro regularization term and assesses pixel-wise distribution differences between known and unknown views.

Finally, we have experimentally validated our conclusions. In sparse view sampling, following the ActiveNeRF protocol, we start with several initial images and supplement new viewpoints to evaluate the information gain brought by our sampling strategy. The experiments demonstrate that our strategy achieves the best performance with the introduction of the same number of new viewpoints. For few-shot novel view synthesis, we compare our designed regularization terms with state-of-the-art baselines, showing consistent improvements across three datasets. An ablation study further analyzes the contribution of each term. Remarkably, the mutual information metric, intuitive and straightforward yet theoretically robust, proves to efficiently guide the NeRF process at both input and output stages with simple quantitative computation in our framework.

## 2 RELATED WORK

**Mutual Information**   Mutual information is a basic concept in information theory and it has many applications in machine learning. Oord et al. (2018) starts the research for unsupervised representation learning train feature extractors by maximizing an estimate of the mutual information (MI) between different views of the data. This work has been expanded in various directions, including the explanation of this principle (Tschannen et al., 2019), the experiments improvement in more datasets (Henaff, 2020), and the application of contrastive learning to the multiview setting (Tian et al., 2020). While the primary focus of their work is on unsupervised learning tasks, our research is centered on supervised learning with sparse samples. However, the concept of leveraging information from unlabeled data is also adopted in our approach.

**Active Learning**   Active learning (Settles, 2009) is a special case of machine learning in which a learning algorithm can actively seek user (or another information source) input to label new data points with desired outputs. It has been extensively explored across diverse computer vision tasks (Yi et al., 2016; Sener & Savarese, 2017; Fu et al., 2018; Zolfaghari Bengar et al., 2019). ActiveNeRF (Pan et al., 2022) is the first approach to incorporate an active learning scheme into the NeRF optimization pipeline. We adopt this active learning pipeline for sparse view sampling. In contrast to ActiveNeRF, which primarily focuses on modeling information gain through uncertainty reduction, our approach explores mutual information from both macro and micro perspectives.

**Few-shot Novel View Synthesis**   NeRF (Mildenhall et al., 2020) has become one of the most important methods for synthesizing new viewpoints in 3D scenes (Xiangli et al., 2021; Fridovich-Keil et al., 2022; Takikawa et al., 2021; Yu et al., 2021a; Tancik et al., 2022; Hedman et al., 2021). A growing number of recent works have studied few-shot novel view synthesis via NeRF (Wang et al., 2021; Martin-Brualla et al., 2021; Meng et al., 2021; Kim et al., 2022; Deng et al., 2022; Wang et al., 2023). First, diffusion-model-based methods use generative inference as supplementary information. SparseFusion (Zhou & Tulsiani, 2022) distills a 3D consistent scene representation from a view-conditioned latent diffusion model. Second, some methods additionally extrapolate the scene's geometry and appearance to a new viewpoint. DietNeRF (Jain et al., 2021) introduces semantic consistency loss between observed and unseen views. Third, some methods utilize regularization terms to avoid overfitting and introduce prior knowledge. RegNeRF (Niemeyer et al., 2022) regularizes the geometry and appearance of patches from unobserved viewpoints. FreeNeRF (Yang et al., 2023) proposes to regularize the input frequency range. However, many methods lack a unified theoretical foundation, making it challenging to provide a comprehensive explanation or optimize better. Our goal is to propose a generic framework with interpretable metrics to address this gap.

## 3 SETUP

First, we briefly overview the Neural Radiance Fields (NeRF) framework with key implementation details. NeRF models the 3D scene as a continuous function $F_\theta$, which is discerned through a multi-layer perceptron (MLP).

**Figure 2: The overview of our framework.** First, we leverage mutual information and relative information to quantify the uncertainty in inferring unknown images conditioned on known ones. This involves decomposing the uncertainty into semantic space distance (macro) and pixel space distance (micro). These distances are converted into specific types tailored for quantifying mutual information in different scenarios. In sparse view sampling, a greedy algorithm is employed as a near-optimal solution to minimize mutual information. We use Euclidean distance of camera positions to represent pixel space distance and propose a sequential method that prioritizes either semantics or pixels(shown in the figure). For few-shot view synthesis, we use color distance to represent pixel space distance and maximize mutual information as efficient plug-and-play regularization terms.

Specifically, given a spatial coordinate $\mathbf{x} \in R^3$ in the scene, and a specific observation direction $\mathbf{d} \in R^2$, NeRF is capable of inferring the corresponding RGB color $c$ and a discrete volume density $\sigma$:

$$F_\theta : (\mathbf{x}, \mathbf{d}) \mapsto (c, \sigma).$$

NeRF models are trained based on a classic differentiable volume rendering operation, which establishes the resulting color of any ray passing through the scene volume and projected onto a camera system. Each ray $\mathbf{r}(t) = \mathbf{o} + t\mathbf{d}$ with $t \in \mathbb{R}^+$, determined by the position of camera $\mathbf{o} \in R^3$ and the direction of ray $\mathbf{d}$. Note that for each $t$, $\mathbf{r}(t)$ represents a position in $R^3$. The value of $\sigma$ defines the geometry of the scene and is learned exclusively from this position. However, the value of $\mathbf{c}$ is also dependent on the viewing direction $\mathbf{d}$. Therefore, we have the volume rendering equation as follows to represent the color on the ray $C(\mathbf{r})$:

$$C(\mathbf{r}) = \int_{t_n}^{t_f} T(t)\sigma(\mathbf{r}(t))c(\mathbf{r}(t), \mathbf{d})dt, \quad T(t) = \exp\left(-\int_{t_n}^{t} \sigma(\mathbf{r}(s))ds\right).$$

Given some images with observing direction $\mathbf{d}$ and camera position $\mathbf{o}$, we can get the ground truth color $C(\mathbf{r})$ on the ray. To estimate it, we can use the NeRF and volume rendering equation to calculate $\hat{C}(\mathbf{r})$. To bypass the challenge of computing the continuous integral, it is common to employ a discretization method: randomly sample $N$ time $\{t_1, t_2, \ldots, t_N\}$ and get the position $\{\mathbf{x}_1, \mathbf{x}_2, \ldots, \mathbf{x}_N\}$ on the ray with $\mathbf{x}_i = \mathbf{o} + t_i\mathbf{d}$. Then we can estimate the color by the following equation, where we denote the sampling interval $\delta_i = t_{i+1} - t_i$:

$$\hat{C}(\mathbf{r}) = \sum_{i=1}^{N} T_i \alpha_i \mathbf{c}_i, \quad T_i = \exp(-\sum_{j=1}^{i-1} \sigma_j \delta_j), \quad \alpha_i = 1 - \exp(-\sigma_i \delta_i).$$

Following this volume rendering logic, the NeRF function $F$ is optimized by minimizing the squared error between the estimated color and the real colors of a batch of rays $\mathcal{R}$ that project onto a set of training views of the scene taken from different viewpoints:

$$L_{\text{NeRF}} = \sum_{\mathbf{r} \in \mathcal{R}} \left\| \hat{C}(\mathbf{r}) - C(\mathbf{r}) \right\|^2.$$

While NeRF achieves outstanding results in view synthesis, it traditionally demands a substantial collection of densely captured, camera-calibrated images. Addressing the difficulties of such extensive data collection, we will introduce a more efficient framework in the next section.

## 4 FRAMEWORK

In this section, we outline our principal framework for the algorithm's design. As we need to choose training images instead of rays, we denote $\mathcal{R}$ as the set of images in this section. Given the limited number of training samples, it's essential to select a sparse but information-rich subset, $\mathcal{R}_s \subset \mathcal{R}$, to capture various details of scenes and generalization well in other views of the scenes or object. Therefore, to establish a criterion for assessing the adequacy of an image in capturing scene information, we draw upon principles from information theory to devise an appropriate metric.

In the domain of information theory, mutual information quantifies the reduction in uncertainty of one variable given the knowledge of another. This concept aligns with our objectives in the context of NeRF. Specifically, we utilize the information from a known image, $R$, which includes a subset of views, to infer properties about an unknown image, $\overline{R}$.

**Definition 1** (Mutual Information). Mutual information measures dependencies between random variables. Given two random variables $R$ and $\overline{R}$, it can be understood as how much knowing $R$ reduces the uncertainty in $\overline{R}$ or vice versa. Formally, the mutual information between $R$ and $\overline{R}$ is:

$$I(R, \overline{R}) = H(R) - H(R|\overline{R}) = H(\overline{R}) - H(\overline{R}|R).$$

where $H(R)$ represents the information of the random variables $R$, $H(R|\overline{R})$ represents the relative uncertainty to infer $R$ if we know $\overline{R}$.

In the context of NeRF, $H(R)$ represents the information of the image $R$, and $H(R|\overline{R})$ represents the relative uncertainty to infer unknown $R$ based on known image $\overline{R}$. Our objective is to quantify the mutual information $I(R, \overline{R})$ and deduce information about one image from another to a certain degree. Assuming symmetry among all images and an equal number of rays, we reasonably hypothesize that the inherent information content of each image $H(R)$ is equal. Consequently, we aim to maximize the conditional information $H(R|\overline{R})$ and $H(\overline{R}|R)$.

We then adopt both macro and micro perspectives to describe the conditional information $H(R|\overline{R})$.

From the macro perspective, the semantic features of the entire image serve as indicators of the uncertainty in the relative information. To gauge the similarity between two images, we consider employing the CLIP method, as proposed by Radford et al. (2021) to extract semantic features.

**Definition 2** (Semantic Space Distance). Suppose we have a clip function $f$, we define the semantic space distance between images $R$ and $\overline{R}$ as the 1 - cosine similarity:

$$s(R, \overline{R}) = 1 - \frac{f(R)f(\overline{R})}{\|f(R)\|\|f(\overline{R})\|}.$$

From the micro perspective, we know that we can decompose the relative information between images into the relative difference of the rays. Suppose the rays in the two images can be described as $\mathbf{r}(t) = \mathbf{o} + t\mathbf{d}$ and $\overline{\mathbf{r}}(t) = \overline{\mathbf{o}} + t\overline{\mathbf{d}}$. The direction can be represent as $\mathbf{d} : (\theta_1, \phi_1)$ and $\overline{\mathbf{d}} : (\theta_2, \phi_2)$, $\theta_1, \theta_2 \in U(\theta, \overline{\theta})$ and $\phi_1, \phi_2 \in U(0, 2\pi)$ are sampled from uniform distribution where $\theta$ and $\overline{\theta}$ are fixed parameter. We assume the distance moving in direction $\mathbf{d}$ of two rays are $T_1$ and $T_2$. Then we define the distance between two rays as the combination of Euclidean distance in expectation between the combination of points in these two rays:

**Definition 3** (Pixel Space Distance). We define the distance between images in pixel space as the distance between any two points of rays in these images in expectation:

$$d(R, \overline{R}) = E_{\mathbf{r} \in R, \overline{\mathbf{r}} \in \overline{R}} \left[ \int_0^{T_1} \int_0^{T_2} \|\mathbf{r}(t_1) - \overline{\mathbf{r}}(t_2)\|_2^2 dt_2 dt_1 \right].$$

Note that measuring the distance between images is consistent with measuring the distance between camera positions $\|\mathbf{o} - \overline{\mathbf{o}}\|_2^2$ corresponding to these images by the following lemma:

**Lemma 1.** *Then the distance between two images can be represented by the Euclidean distance of two positions of cameras, $\|\mathbf{o} - \overline{\mathbf{o}}\|_2^2$, by the following equation:*

$$d(R, \overline{R}) = T_1 T_2 \|\mathbf{o} - \overline{\mathbf{o}}\|_2^2 + C,$$

where $C$ is a constant independent of $\mathbf{o}$ and $\overline{\mathbf{o}}$. Therefore, we use the measure $d(R, \overline{R})$ and $s(R, \overline{R})$ to represent the relative information $H(R|\overline{R})$. We make the following assumption:

**Assumption 1.** We assume the relative information of two images $H(R|\overline{R})$ is proportional to the similarity measure and distance measure between two images. That is,

$$H(R|\overline{R}) \propto s(R, \overline{R}), \quad H(R|\overline{R}) \propto d(R, \overline{R}).$$

Note that when we are predicting the information of an uncaptured image $\overline{R}$, we are not limited to using information from a single image in the training set. Rather, we can harness the collective information from multiple images, denoted $R_1, R_2, \ldots R_m$. it becomes necessary to extend the definition of mutual information to encompass multiple variables, capturing the interdependencies among more than two variables. Drawing on insights from (Williams & Beer, 2010), we understand that the mutual information across multiple images can be broken down into the maximal mutual information observed between any two images. The formulation is as detailed below:

**Definition 4** (Mutual Information for multiple images)**.** Suppose we have several images $R_1, R_2, \ldots R_m$ in the training set. Then we want to infer the information of an unknown image $\overline{R}$, the mutual information of this image corresponding to other images is defined as:

$$I(R_1, R_2, \ldots R_m; \overline{R}) = \max_{i=1,2,\ldots m} I(R_i, \overline{R}).$$

After presenting the framework, we will illustrate our algorithm's efficacy in addressing two critical tasks which detailed in the subsequent sections: sparse view sampling and few-shot view synthesis.

## 5 SPARSE VIEW SAMPLING

Sparse view sampling, proposed by ActiveNeRF (Pan et al., 2022), is an active learning scheme designed to enhance the quality of NeRF by strategically selecting additional viewpoints. In this setting, we begin with a limited number of training images, and a candidate set of viewpoints for which we **do not possess the corresponding ground truth images**. **It is only after a viewpoint is selected that we acquire its ground truth image**, subsequently transferring it from the candidate to the training set. By analyzing the shortcomings of initial images, we strategically select additional viewpoints and then get the corresponding images to improve the NeRF model's synthesis quality. For instance, if constrained to capture only three images of the Eiffel Tower, we are presented with various potential viewpoints from the sky or ground. Sparse view sampling involves selecting the most informative viewpoints based on the initial images.

Our framework selects an informative subset of views by **minimizing mutual information** without knowing the ground truth images beforehand. It stems from the observation that lower mutual information reflects reduced redundancy between views. For example, highly similar images exhibit high mutual information, indicating redundancy if both are selected. We aim to design an algorithm that intelligently chooses images based solely on the existing images and the candidate view positions.

First, let's consider a global optimization problem. Suppose the whole set of images is $\mathcal{R}$ and we need to choose the subset of images $\mathcal{R}_s$. We represent $R_{i \neq j}$ as all the images in $\mathcal{R}$ without the image $R_j$, then our goal can be formally described as minimizing the mutual information for $R_{i \neq j}$ and $R_j$. By Definition 4, it can be represent as the maximal mutual information between $R_i$ and $R_j$:

$$\min_{\mathcal{R}_s \subset \mathcal{R}} \max_{R_j \in \mathcal{R}_s} I(R_{i \neq j}; R_j) = \min_{\mathcal{R}_s \subset \mathcal{R}} \max_{R_i, R_j \in \mathcal{R}_s} I(R_i, R_j).$$

Given $N$ figures in the subset $\mathcal{R}_s$, we can reformulate the goal from minimizing mutual information to maximizing relative information between images by Definition 1. Thus, the problem becomes:

$$\max_{\mathcal{R}_s \subset \mathcal{R}} \delta \quad \text{s.t. } H(R_i|R_j) \geq \delta, \forall i, j \in \{1, 2, \ldots N\}, i \neq j.$$

Then we use the solution as the training images to construct an informative NeRF.

### 5.1 GREEDY ALGORITHM

Solving this problem is challenging without initial ground truth images for all candidate viewpoints and involves balancing $O(N^2)$ constraints. Thus, we adopt a near-optimal approximation algorithm

| Sampling Strategies | Setting I, 20 observations: | | | Setting II, 10 observations: | | |
|---|---|---|---|---|---|---|
| | PSNR ↑ | SSIM ↑ | LPIPS ↓ | PSNR ↑ | SSIM ↑ | LPIPS ↓ |
| NeRF + Rand | 16.626 | 0.822 | 0.186 | 15.111 | 0.779 | 0.256 |
| NeRF + FVS(**Pixel**) | 17.832 | 0.819 | 0.186 | 15.723 | 0.787 | 0.227 |
| NeRF + **Semantic** | 17.334 | 0.833 | 0.171 | 15.472 | 0.795 | 0.219 |
| ActiveNeRF | 18.732 | 0.826 | 0.181 | 16.353 | 0.792 | 0.226 |
| Ours (S→P) | 18.930 | **0.846** | **0.149** | 16.718 | **0.810** | **0.205** |
| Ours (P→S) | **20.093** | 0.841 | 0.162 | **17.314** | 0.801 | 0.209 |

**Table 1: Quantitative comparison in Active Learning settings on Blender. NeRF + Rand:** Randomly capture new views in the candidates. **NeRF + FVS(Pixel):** Capture new views using furthest view sampling to maximize pixel space distance. **NeRF + Semantic:** Capture new views using CLIP to maximize semantic space distance. **ActiveNeRF:** Capture new views using the ActiveNeRF scheme. **Ours (S→P):** First choose 20 views with the highest semantic space distance, then capture views within them based on the highest pixel space distance (camera pose). **Ours (P→S):** First capture 20 views with the highest pixel space distance, then capture views within them based on semantic space distance. **Setting I:** 4 initial observations and 4 extra observations obtained at 40K,80K,120K and 160K iterations. **Setting II:** 2 initial observations and 2 extra observations obtained at 40K,80K,120K and 160K iterations. 200K iterations for training in total. All results are produced using the ActiveNeRF codebase.

that is both tractable and computationally efficient. We use a look-ahead strategy and greedy method to select views. Over $N$ iterations, we choose an image in each iteration that has minimal information overlap with the already selected images. In the $i$-th iteration, we solve the following problem:

$$\max_{R_i \in \mathcal{R}} \delta_i \ \text{ s.t. } H(R_i|R_j) \geq \delta_i, \forall 1 \leq j < i \,.$$

Then the mutual information of $N$ images we choose is $\tilde{\delta} = \min\{\delta_1, \delta_2, \ldots, \delta_N\}$. Although this algorithm can not achieve the global minimum point of the primal problem, it is a 2-approximation based on the following lemma:

**Lemma 2.** *Assume the optimal value of the primal problem is $\delta$, the value we achieved by the greedy algorithm is $\tilde{\delta}$, then we have $\tilde{\delta} \geq \frac{1}{2}\delta$.*

This lemma ensures that our greedy algorithm provides a good approximation to the optimal solution. Additionally, our algorithm substantially reduces the computational cost of the problem, as we only have $O(N)$ constraints in each instance, as opposed to $O(N^2)$. We will subsequently employ this iterative strategy for image selection in our experiments.

## 5.2 EXPERIMENTS

**Setup** Our greedy algorithm in Section 5.1 follows a 'train-render-evaluate-pick' scheme similar to that in Active Learning (Pan et al., 2022): 1) start by training a NeRF model with initial observations, 2) render images from candidate views and evaluate them to select valuable ones, 3) train the NeRF model with the newly acquired ground-truth images corresponding to these selected views, then repeat to step 2. Compared to ActiveNeRF, we modify the evaluation metric in step 2 as minimizing mutual information, considering both semantic space distance and pixel space distance discussed in Section 4.

**Design** By Assumption 1 and Lemma 1, we identify a viewpoint that exhibits both low semantic similarity measured by CLIP (Radford et al., 2021) (large semantic space distance) and a considerable distance in camera positions (large pixel space distance). If we consider only camera pose, furthest view sampling (FVS) is optimal. However, incorporating semantic constraints necessitates balancing these two criteria. We propose a sequential approach: first prioritize semantics to select a subset from candidates, then evaluate based on camera pose (S→P), or vice versa (P→S). This strategy navigates the tradeoff without a tricky balance hyperparameter. The technical appendix provides more discussions.

**Dataset Dataset and Metric** We extensively evaluate our approach on the Blender (Mildenhall et al., 2020) dataset, which contains 8 synthetic objects with complex geometry and realistic materials and is classical in the NeRF research. We report the image quality metrics PSNR, SSIM, and LPIPS for

| Sampling Strategies | Setting I, 20 observations: | | |
|---|---|---|---|
| | PSNR ↑ | SSIM ↑ | LPIPS ↓ |
| semantic distance + 0.1 * pixel distance | 18.781 | 0.833 | 0.153 |
| semantic distance + pixel distance | 19.266 | 0.837 | 0.159 |
| semantic distance + 10 * pixel distance | 18.345 | 0.821 | 0.187 |
| Ours (S→P) | 18.930 | **0.846** | **0.149** |
| Ours (P→S) | **20.093** | 0.841 | 0.162 |

**Table 2: Ablations on balancing two metrics.** We introduce hyperparameters to balance pixel and semantic space distances, considering both factors simultaneously. Our sequential approaches (**Ours (S→P)** or **Ours (P→S)**) outperform the alternatives.

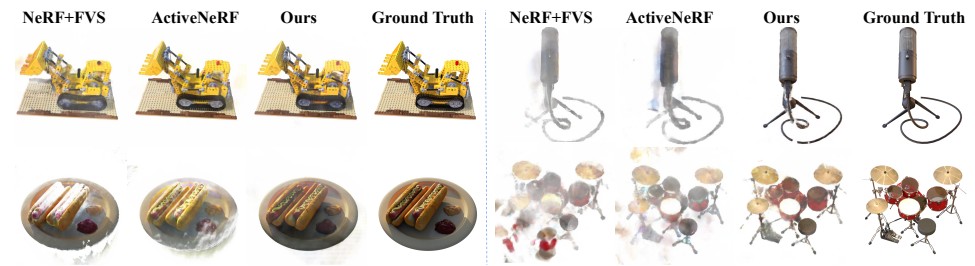

**Figure 3: Quantitative comparison in Active Learning settings on Blender.** Given limited input views, our strategy can select better candidate views. Our rendered images without excessively blurry boundaries exhibit greater clarity compared to those rendered by ActiveNeRF.

evaluations. SSIM measures differences in luminance, contrast, and structure, focusing on perceptual properties. PSNR assesses the absolute error between pixels, emphasizing pixel-wise comparison in a micro way. LPIPS quantifies perceptual similarity, capturing more global visual differences in a macro way.

**Results** We demonstrate the performance of our sampling strategy on the Blender dataset compared to baseline approaches in Table 1 and Figure 3. Our strategy outperforms baselines in view synthesis quality. Our method, which considers both the semantic space distance between visible and invisible views and a tendency towards uniform sampling, provides better sampling guidance under a limited input budget. When prioritizing semantic space distance before pixel space distance (*Ours (S→P)*), we observe lower LPIPS scores (-17.6%/-9.2%) and higher SSIM scores (+2.4%/+2.3%), aligning more closely with human perception. Conversely, prioritizing pixel space distance first (*Ours (P→S)*) yields higher PSNR scores (+7.3%/+5.9%), reflecting differences in raw pixel values. In addition, as shown in Table 2, our sequential method can get better results than simultaneous method.

**Ablation** We conduct ablation studies using only semantic space distance or only pixel distance. As shown by *NeRF + FVS(Pixel)* and *NeRF + Semantic* in Table 1, considering either factor improves performance compared to the naive method. However, combining both metrics yields even better results, as seen in *Ours (S→P)* and *Ours (P→S)*.

## 6 FEW-SHOT VIEW SYNTHESIS

In this section, we address the challenge of few-shot view synthesis, which is more prevalent in NeRF research: optimizing the NeRF model with limited and fixed training images. The key is to extract valuable information from the training set while maintaining generalization capabilities.

A natural approach involves randomly rendering images from NeRF that lack ground truth and leveraging the information extracted from them. Based on this, our objective is to **maximize mutual information** between visible training images and invisible inferred images, expecting that inferred images without ground truth can gain more relevant information from known images.

To tackle this, we introduce two regularization terms to train a generalizable NeRF model.

| Method | DTU(Object) | | DTU(Full image) | | LLFF | | |
|---|---|---|---|---|---|---|---|
| | PSNR ↑ | SSIM ↑ | PSNR ↑ | SSIM ↑ | PSNR ↑ | SSIM ↑ | LPIPS ↓ |
| Mip-NeRF | 9.10 | 0.578 | 7.94 | 0.235 | 16.11 | 0.401 | 0.460 |
| DiffusioNeRF | 16.20 | 0.698 | / | / | 19.79 | 0.568 | 0.338 |
| DietNeRF | 11.85 | 0.633 | 10.01 | 0.354 | 14.94 | 0.370 | 0.496 |
| **DietNeRF+Ours** | 13.04 | 0.711 | 11.95 | 0.410 | 16.01 | 0.433 | 0.443 |
| RegNeRF | 18.50 | 0.744 | 15.00 | 0.606 | 18.84 | 0.573 | 0.345 |
| **RegNeRF+Ours** | 19.78 | 0.791 | 15.79 | 0.634 | 19.44 | 0.611 | 0.322 |
| FreeNeRF | 19.92 | 0.787 | 18.02 | 0.680 | 19.63 | 0.612 | 0.308 |
| **FreeNeRF+Ours** | **20.42** (+0.50) | **0.814** (+0.027) | **18.63** (+0.61) | **0.712** (+0.032) | **20.17** (+0.54) | **0.634** (+0.022) | **0.274** (-0.034) |

Table 3: **Quantitative comparison on LLFF and DTU.** There are 3 input views for training, consistent with FreeNeRF. On DTU, we use objects' masks to remove the background when computing metrics, as full-image evaluation is biased towards the background, as reported by (Yu et al., 2021b; Niemeyer et al., 2022).

### 6.1 THE DESIGN OF REGULARIZATION TERM

In our framework, maximizing the mutual information between images involves minimizing both semantic space distance and pixel space distance. For the former, we can use CLIP (Radford et al., 2021) as a macro regularization. However, camera position cannot be used to analyze pixel space distance as in Section 5 because it is independent of NeRF parameters and cannot be optimized. Thus, we need a new metric that depends on both camera position and network parameters to serve as the micro regularization.

To fully utilize simple pixel-wise information, we establish a close relationship between the difference in RGB color and the distance of the camera position, detailed in the following lemma:

**Lemma 3.** *Assume we have two rays* $\mathbf{r}(t) = \mathbf{o} + t\mathbf{d}$ *and* $\overline{\mathbf{r}}(t) = \overline{\mathbf{o}} + t\overline{\mathbf{d}}$. *Assume the function* $\sigma(\mathbf{r}(t))$ *and* $c(\mathbf{r}(t), \mathbf{d})$ *learned by MLP is L-Lipschitz of* $\mathbf{r}(t)$ *and* $\mathbf{d}$(*We usually use Relu activation in MLP and it is a Lipschitz function*). *Then the distance between RGB colors of two rays can be upper bounded by the Euclidean distance of two positions of cameras,* $\|\mathbf{o} - \overline{\mathbf{o}}\|$, *and it can be represented as*

$$\|\hat{C}(\mathbf{r}) - \hat{C}(\overline{\mathbf{r}})\| \leq 3L\|\mathbf{o} - \overline{\mathbf{o}}\| + C.$$

*where* $C$ *represent a constant independent of the distance* $\|\mathbf{o} - \overline{\mathbf{o}}\|$.

From Lemma 3, we know that the difference in RGB color serves as a lower bound for the difference in camera position. According to Lemma 1, it also acts as a lower bound for pixel space distance. Therefore, to reduce pixel space distance, we aim to minimize the color difference (like color variance or KL divergence) between training ground truth images and randomly rendered images.

Then we can define our two plug-and-play regularization terms added to the loss function:

$$L_{\text{macro}}(R, \overline{R}) = s(R, \overline{R}) = 1 - \frac{f(R)f(\overline{R})}{\|f(R)\|\|f(\overline{R})\|},$$

$$L_{\text{micro}}(R, \overline{R}) = \sum_{\mathbf{r} \in R, \overline{\mathbf{r}} \in \overline{R}} \|\hat{C}(\mathbf{r}) - \hat{C}(\overline{\mathbf{r}})\|.$$

### 6.2 EXPERIMENTS

**Setup** To demonstrate the effectiveness of our method, we evaluate it on three datasets under few-shot settings: the Blender dataset (Mildenhall et al., 2020), the DTU dataset (Jensen et al., 2014), and the LLFF dataset (Mildenhall et al., 2019). We compare our method with classical NeRF and state-of-the-art baselines like FreeNeRF (Yang et al., 2023).

**Design** We add our regularization terms $L_{\text{macro}}$ and $L_{\text{micro}}$ to maximize mutual information. Specifically, $L_{\text{micro}}$ is the variance of the mean color value between training images and randomly rendered images, ensuring that the color difference is constrained to some degree.

| Method | PSNR ↑ | SSIM ↑ | LPIPS ↓ |
|---|---|---|---|
| NeRF | 14.934 | 0.687 | 0.318 |
| NV | 17.859 | 0.741 | 0.245 |
| Simplified NeRF | 20.092 | 0.822 | 0.179 |
| NeRF + $L_{micro}$ | 20.101 (+5.167) | 0.799(+0.112) | 0.151(-0.167) |
| NeRF + $L_{macro}$ (DietNeRF) | 22.503 (+7.569) | 0.823 (+0.136) | 0.124(-0.194) |
| **NeRF + $L_{micro}$ + $L_{macro}$ (Ours)** | **23.394 (+8.460)** | **0.859 (+0.172)** | **0.103 (-0.215)** |
| FreeNeRF | 24.259 | 0.883 | 0.098 |
| **FreeNeRF+Ours** | **24.896 (+0.637)** | **0.904 (+0.021)** | **0.086 (-0.012)** |

**Table 4: Quantitative comparison on Blender.** There are 8 input views for training, consistent with FreeNeRF. For DietNeRF, the consistency loss actually belongs to the $L_{\text{macro}}$, so DietNeRF is a degradation of our framework.

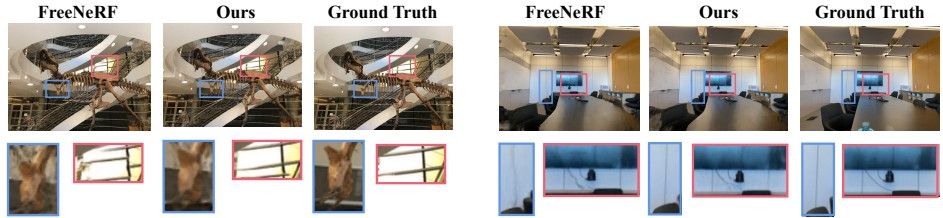

**Figure 4: Qualitative comparison on LLFF.** Given 3 input views, we show novel views rendered by FreeNeRF and ours Compared with FreeNeRF. FreeNeRF fails to render sharp outlines in some places, but our additional losses can gain a more detailed skeleton structure and better geometry for the observed objects.

**Comparison with baseline methods.** Table 3 and Figure 4 present the quantitative and qualitative results of the DTU dataset and the LLFF dataset under a 3-view setting. Table 4 also presents the improvements on the blender dataset under an 8-view setting. Incorporating $L_{\text{macro}}$ and $L_{\text{micro}}$, our method builds on the RegNeRF/FreeNeRF framework, introducing additional regularization terms. These constraints enhance the consistency of unobservable views from both semantic and color perspectives. The improvements in results validate the effectiveness of our approach. Our framework facilitates the design and application of various regularization terms, leading to improved outcomes. While we focused on $L_{\text{macro}}$ and $L_{\text{micro}}$, our framework is not limited to these specific terms. It allows for the exploration of various regularization methods, providing flexibility to experiment with and integrate different approaches. Detailed explanations are provided in the appendix.

**Ablations.** In Table 4, we decompose two regularization terms to prove the effectiveness of each. For clearer comparison, we compare with classical NeRF, as many methods, such as DietNeRF (Jain et al., 2021) with semantic consistency loss or FreeNeRF (Yang et al., 2023) with free frequency regularizations, include various regularization terms that may overlap with ours to some extent. If we normalize the improvements in PSNR, SSIM, and LPIPS with both regularization terms to 1, the improvements with only $L_{\text{micro}}$ are 0.61, 0.65, and 0.78, respectively. With only $L_{\text{macro}}$, the improvements are 0.89, 0.79, and 0.90. While $L_{\text{macro}}$ has a slightly more significant impact, using both terms together yields the best results.

## 7 Conclusion, Limitations and Future Directions

This paper presents a novel NeRF framework under limited samples using Mutual Information Theory. We introduce mutual information from both macro (semantic space) and micro (pixel space) perspectives in different settings. In sparse view sampling, we employ a greedy algorithm to minimize mutual information. In few-shot view synthesis, we utilize plug-and-play regularization terms to maximize it. Experiments across different settings validate the robustness of our framework.

Our framework has some limitations, particularly in terms of comparisons with the diffusion-based methods. We were unable to include these comparisons due to the lack of open-source code or differing dataset settings, which are detailed in the appendix. Future work should aim to incorporate more baseline methods and explore additional variations within our framework.

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

## A PROOF

In this section, we mainly prove the lemmas used in our paper.

### A.1 PROOF OF LEMMA 1

**Lemma 1.** *Then the distance between two images can be represented by the Euclidean distance of two positions of cameras, $\|\mathbf{o} - \overline{\mathbf{o}}\|_2^2$, by the following equation:*

$$d(R, \overline{R}) = T_1 T_2 \|\mathbf{o} - \overline{\mathbf{o}}\|_2^2 + C \,,$$

*Proof.* Denote $\mathbf{o} = (o_1, o_2, o_3), \overline{\mathbf{o}} = (\overline{o}_1, \overline{o}_2, \overline{o}_3)$. Using the property of uniform distribution and the spherical polar coordinates, we have

$$d(R, \overline{R}) = E_{\mathbf{r} \in R, \overline{\mathbf{r}} \in \overline{R}} \left[ \int_0^{T_1} \int_0^{T_2} \|\mathbf{r}(t_1) - \overline{\mathbf{r}}(t_2)\|_2^2 dt_2 dt_1 \right]$$

$$= \int_0^{T_1} \int_0^{T_2} E_{\theta, \phi}[(o_1 - \overline{o}_1 + t_1 \cos\theta_1 \cos\phi_1 - t_2 \cos\theta_2 \cos\phi_2)^2$$

$$+ (o_2 - \overline{o}_2 + t_1 \cos\theta_1 \sin\phi_1 - t_2 \cos\theta_2 \sin\phi_2)^2 + (o_3 - \overline{o}_3 + t_1 \sin\theta_1 - t_2 \sin\theta_2)^2] dt_2 dt_1$$

$$= \int_0^{T_1} \int_0^{T_2} [\|\mathbf{o} - \overline{\mathbf{o}}\|_2^2 + C_1(\mathbf{o}, \overline{\mathbf{o}}, t_1, t_2) + C_2(t_1, t_2)] dt_2 dt_1$$

$$= T_1 T_2 \|\mathbf{o} - \overline{\mathbf{o}}\|_2^2 + C + \int_0^{T_1} \int_0^{T_2} C_1(\mathbf{o}, \overline{\mathbf{o}}, t_1, t_2) dt_2 dt_1 \,.$$

where we represent

$$C_1(\mathbf{o}, \overline{\mathbf{o}}, t_1, t_2) = 2(o_1 - \overline{o}_1) E_{\theta, \phi}[t_1 \cos\theta_1 \cos\phi_1 - t_2 \cos\theta_2 \cos\phi_2]$$
$$+ 2(o_2 - \overline{o}_2) E_{\theta, \phi}[t_1 \cos\theta_1 \sin\phi_1 - t_2 \cos\theta_2 \sin\phi_2] + 2(o_3 - \overline{o}_3) E_{\theta, \phi}[t_1 \sin\theta_1 - t_2 \sin\theta_2] \,.$$

and let $C_2(t_1, t_2)$ include all items that are not related to $\mathbf{o}$ and $\overline{\mathbf{o}}$.

By the symmetry property of $\phi \in U(0, 2\pi)$, we know that the $E_{\theta, \phi}[\sin\phi] = E_{\theta, \phi}[\cos\phi] = 0$. Furthermore, by the i.i.d property of $\theta_1$ and $\theta_2$, we have $E_{\theta, \phi}[\sin\theta_1] = E_{\theta, \phi}[\sin\theta_2]$. Observing that the integration over $t_1$ and $t_2$ is also symmetrical, we can deduce that $\int_0^{T_1} \int_0^{T_2} C_1(\mathbf{o}, \overline{\mathbf{o}}, t_1, t_2) dt_2 dt_1 = 0$. Therefore, we finally get

$$d(R, \overline{R}) = T_1 T_2 \|\mathbf{o} - \overline{\mathbf{o}}\|_2^2 + C \,.$$

where $C = \int_0^{T_1} \int_0^{T_2} C_2(t_1, t_2) dt_2 dt_1$ represent a constant independent of the camera position $\mathbf{o}$ and $\overline{\mathbf{o}}$. $\square$

### A.2 PROOF OF LEMMA 2

**Lemma 2.** *Assume the optimal value of the primal problem is $\delta$, the value we achieved by the greedy algorithm is $\tilde{\delta}$, then we have $\tilde{\delta} \geq \frac{1}{2}\delta$.*

*Proof.* Suppose the optimal solution in the primal problem is $R_1, R_2, \ldots, R_N$, the optimal solution obtained by our greedy algorithm is $\tilde{R}_1, \tilde{R}_2, \ldots, \tilde{R}_N$.

We first prove that the optimal value in the $i + 1$-th iteration of our method is not larger than the optimal value in the $i$-th iteration. Assume not, $\delta_{i+1} > \delta_i$, then we can find the image $\tilde{R}_{i+1}$ satisfy $H(\tilde{R}_{i+1}|R_j) \geq \delta_{i+1}$ for all $j \leq i$. Because in the $i$-th iteration we only have the constraints $H(\tilde{R}|\tilde{R}_j) \geq \delta_i$ for all $j \leq i - 1$, therefore, we take $\tilde{R} = \tilde{R}_{i+1}$ will satisfy this constraints, with the value $\delta_{i+1} > \delta_i$, contradict with the property that $\delta_i$ is the optimal solution in the $i$-th iteration. So the optimal value in the $i + 1$-th iteration of our method is not larger than the optimal value in the $i$-th iteration.

Then we can assume the optimal value we find in each iteration as $\delta_1 \geq \delta_2 \geq \ldots \geq \delta_N$. So we have $\tilde{\delta} = \min\{\delta_1, \delta_2, \ldots, \delta_N\} = \delta_N$.

**Then we prove the conclusion by contradiction.** Suppose we have $\tilde{\delta} < \frac{1}{2}\delta$. Assume we have $n$ common images of the solution of the primal problem and the solution obtained by our greedy algorithm. By the solution $\tilde{\delta} < \frac{1}{2}\delta$ we know that $n <= N - 1$. So there are $N - n$ images in the primal solution that do not appear in our solution. Suppose the different images of primal solution are $R_{i_1}, R_{i_2}, \ldots, R_{i_{N-n}}$ and the different images in our solution are $\tilde{R}_{i_1}, \tilde{R}_{i_2}, \ldots, \tilde{R}_{i_{N-n}}$. Then we consider the optimization problem in the iteration that we choose the last different image. Then we consider the last iteration of our algorithm:

$$\max_{R \in \mathcal{R}} \delta_N \text{ s.t. } H(R|\tilde{R}_j) \geq \delta_N, \forall 1 \leq j \leq N - 1.$$

Then for the different images, $R_{i_1}, R_{i_2}, \ldots, R_{i_{N-n}}$ appear in primal solution but do not appear in our solution, we have that taking these images in the solution will incur a smaller solution. That is, for each $R$ in $R_{i_1}, R_{i_2}, \ldots, R_{i_{N-n}}$, we have a corresponding image $\tilde{R}$ in $\tilde{R}_1, \ldots, \tilde{R}_{N-1}$, incur the relative difference $H(R|\tilde{R}) \leq \delta_N = \tilde{\delta}$. By the definition of $\delta$, we know that $\tilde{R}$ can only choose in the difference set $\tilde{R}_{i_1}, \tilde{R}_{i_2}, \ldots, \tilde{R}_{i_{N-n}}$. Then we consider two cases:

- Case 1. **The optimal solution of the last iteration $\tilde{R}_N$ is not in the set of common image.** In this case, Because we have not selected it in the first $n-1$ iterations, we only have $N-n-1$ images to choose for the corresponding images selected in our algorithm which satisfy $H(R|\tilde{R}) \leq \delta_N = \tilde{\delta}$. However, we have $R_{i_1}, R_{i_2}, \ldots, R_{i_{N-n}}$ in optimal solution of primal set, there are $N - n$ images satisfy this inequality. Therefore, by the Pigeonhole Principle, there exists two images in $R_{i_1}, R_{i_2}, \ldots, R_{i_{N-n}}$ corresponding to the same image $\tilde{R}_{i_k}$ in $\tilde{R}_{i_1}, \tilde{R}_{i_2}, \ldots, \tilde{R}_{i_{N-n}}$ that incur $H(R|\tilde{R}) \leq \tilde{\delta}$. By Assumption 1 we know that $H(R|\overline{R}) \propto d(R, \overline{R})$. By the definition of $d(R, \overline{R}) = E_{\mathbf{r} \in R, \overline{\mathbf{r}} \in \overline{R}} \left[ \int_0^{T_1} \int_0^{T_2} \|\mathbf{r}(t_1) - \overline{\mathbf{r}}(t_2)\|_2^2 dt_2 dt_1 \right]$ we know that it satisfy the triangle inequality: $d(R_{i_1}, R_{i_2}) \leq d(R_{i_1}, \tilde{R}_{i_k}) + d(R_{i_2}, \tilde{R}_{i_k})$. Therefore we can get the triangle inequality of $H$:

  $$H(R_{i_1}|R_{i_2}) \leq H(R_{i_1}|\tilde{R}_{i_k}) + H(R_{i_2}|\tilde{R}_{i_k}) < \frac{\delta}{2} + \frac{\delta}{2} = \delta.$$

  This is contradictory to the fact that these two images are in the solution of the primal problem with distance $H(R_{i_1}|R_{i_2}) \geq \delta$.

- Case 2. **The optimal solution of the last iteration $\tilde{R}_N$ is in the set of common image.** In this case, we have $N - n$ images to choose for the corresponding images selected in our algorithm which satisfy $H(R|\tilde{R}) \leq \delta_N = \tilde{\delta}$. Note that we have $R_{i_1}, R_{i_2}, \ldots, R_{i_{N-n}}$ in optimal solution of primal set, there are $N - n$ images satisfy this inequality. If there are two images in the primal set corresponding to the same image selected by our algorithm, using the analysis of case 1 will get a contradiction. Therefore, we only need to consider the case they are all corresponding to different images in our set, that is, each image $\tilde{R}_{i_k}$ in our set has a unique corresponding image $R_{i_l}$ in the primal set. However, note that the last iteration solution $\tilde{R}_N$ is in the set of common images and it also satisfies the constraint, that is, it also corresponds to an image $R$, satisfy the inequality $H(\tilde{R}_N|R) = \delta_N < \frac{\delta}{2}$. By the definition of $\delta$, we know that $R$ must be in the different sets in our solution, not the common set. But we have proved that each image in $\tilde{R}_{i_1}, \tilde{R}_{i_2}, \ldots, \tilde{R}_{i_{N-n}}$ corresponds to an image in primal set satisfies the inequality. Suppose $H(R|R_{i_k}) < \frac{\delta}{2}$. By Assumption 1 we know that the relative difference is proportional to the distance metric so it also satisfies the triangle inequality. So we have:

  $$H(\tilde{R}_N|R_{i_k}) \leq H(\tilde{R}_N|R) + H(R|R_{i_k}) < \frac{\delta}{2} + \frac{\delta}{2} = \delta.$$

  This is contradictory to the fact that these two images $\tilde{R}_N$ and $R_{i_k}$ are in the solution of the primal problem with distance $H(\tilde{R}_N|R_{i_k}) \geq \delta$.

Therefore, we have proved this lemma by contradiction and show that $\tilde{\delta} \geq \frac{1}{2}\delta$.

$\square$

### A.3 PROOF OF LEMMA 3

**Lemma 3.** *Assume we have two rays $\mathbf{r}(t) = \mathbf{o} + t\mathbf{d}$ and $\overline{\mathbf{r}}(t) = \overline{\mathbf{o}} + t\overline{\mathbf{d}}$. Assume the function $\sigma(\mathbf{r}(t))$ and $c(\mathbf{r}(t), \mathbf{d})$ learned by MLP is $L$-Lipschitz of $\mathbf{r}(t)$ and $\mathbf{d}$(We usually use Relu activation in MLP and it is a Lipschitz function). Then the distance between RGB colors of two rays can be upper bounded by the Euclidean distance of two positions of cameras, $\|\mathbf{o} - \overline{\mathbf{o}}\|$, and it can be represented as*

$$\|\hat{C}(\mathbf{r}) - \hat{C}(\overline{\mathbf{r}})\| \leq 3L\|\mathbf{o} - \overline{\mathbf{o}}\| + C.$$

*where $C$ represent a constant independent of the distance $\|\mathbf{o} - \overline{\mathbf{o}}\|$.*

*Proof.* By the definition of $\hat{C}(\mathbf{r})$, we have

$$\|\hat{C}(\mathbf{r}) - \hat{C}(\overline{\mathbf{r}})\| \leq \sum_{i=1}^{N} \|T_i \alpha_i \mathbf{c}_i - \overline{T}_i \overline{\alpha}_i \overline{\mathbf{c}}_i\|.$$

Then we analysis $|T_i - \overline{T}_i|$, $|\alpha_i - \overline{\alpha}_i|$, $\|\mathbf{c}_i - \overline{\mathbf{c}}_i\|$ separately. We have

$$\begin{aligned}
|T_i - \overline{T}_i| &= |\exp(-\sum_{j=1}^{i-1} \sigma_j \delta_j) - \exp(-\sum_{j=1}^{i-1} \overline{\sigma}_j \delta_j)| \\
&= |\exp(-\sum_{j=1}^{i-1} \sigma(\mathbf{o} + t_j \mathbf{d})\delta_j) - \exp(-\sum_{j=1}^{i-1} \sigma(\overline{\mathbf{o}} + t_j \overline{\mathbf{d}})\delta_j)| \\
&\leq |-\sum_{j=1}^{i-1} \sigma(\mathbf{o} + t_j \mathbf{d})\delta_j + \sum_{j=1}^{i-1} \sigma(\overline{\mathbf{o}} + t_j \overline{\mathbf{d}})\delta_j| \\
&\leq \sum_{j=1}^{i-1} |\sigma(\mathbf{o} + t_j \mathbf{d}) - \sigma(\overline{\mathbf{o}} + t_j \overline{\mathbf{d}})|\delta_j \\
&\leq \sum_{j=1}^{i-1} L\|\mathbf{o} + t_j \mathbf{d} - \overline{\mathbf{o}} + t_j \overline{\mathbf{d}}\|\delta_j \\
&\leq (\sum_{j=1}^{i-1} \delta_j L)\|\mathbf{o} - \overline{\mathbf{o}}\| + (\sum_{j=1}^{i-1} \delta_j t_j L)\|\mathbf{d} - \overline{\mathbf{d}}\| \\
&= t_i L\|\mathbf{o} - \overline{\mathbf{o}}\| + (\sum_{j=1}^{i-1} \delta_j t_j L)\|\mathbf{d} - \overline{\mathbf{d}}\|.
\end{aligned}$$

where the first inequality is because $|e^{-x} - e^{-y}| \leq |x - y|$, the second inequality is by the lipschitz property of $\sigma$, the final equality is because $\delta_i = t_{i+1} - t_i$ and $t_1 = 0$. We also have

$$\begin{aligned}
|\alpha_i - \overline{\alpha}_i| &= |\exp(-\overline{\sigma}_i \delta_i) - \exp(-\sigma_i \delta_i)| \\
&\leq |\sigma(\mathbf{o} + t_i \mathbf{d}) - \sigma(\overline{\mathbf{o}} + t_i \overline{\mathbf{d}})|\delta_i \\
&\leq L\|\mathbf{o} + t_i \mathbf{d} - \overline{\mathbf{o}} + t_i \overline{\mathbf{d}}\|\delta_i \\
&\leq \delta_i L\|\mathbf{o} - \overline{\mathbf{o}}\| + t_i \delta_i L\|\mathbf{d} - \overline{\mathbf{d}}\|.
\end{aligned}$$

Finally, we have

$$\begin{aligned}
\|\mathbf{c}_i - \overline{\mathbf{c}}_i\| &= \|\mathbf{c}(\mathbf{o} + t_i \mathbf{d}, \mathbf{d}) - \mathbf{c}(\overline{\mathbf{o}} + t_i \overline{\mathbf{d}}, \overline{\mathbf{d}})\| \\
&\leq \|\mathbf{c}(\mathbf{o} + t_i \mathbf{d}, \mathbf{d}) - \mathbf{c}(\overline{\mathbf{o}} + t_i \overline{\mathbf{d}}, \mathbf{d})\| + \|\mathbf{c}(\overline{\mathbf{o}} + t_i \overline{\mathbf{d}}, \mathbf{d}) - \mathbf{c}(\overline{\mathbf{o}} + t_i \overline{\mathbf{d}}, \overline{\mathbf{d}})\| \\
&\leq L\|\mathbf{o} - \overline{\mathbf{o}} + t_i(\mathbf{d} - \overline{\mathbf{d}})\| + L\|\mathbf{d} - \overline{\mathbf{d}}\| \\
&\leq L\|\mathbf{o} - \overline{\mathbf{o}}\| + L(t_i + 1)\|\mathbf{d} - \overline{\mathbf{d}}\|.
\end{aligned}$$

By the expression of $T_i$ and $\alpha_i$, we know $T_i \leq 1$, $\alpha_i \leq \delta_i$. As $c_i$ represents the RGB color, the norm of $c_i$ is also bounded by 1. Furthermore, the difference of viewing direction $\|d - \overline{d}\|$ is bounded. Therefore, we finally have the following upper bound:

$$\|\hat{C}(\mathbf{r}) - \hat{C}(\overline{\mathbf{r}})\| \leq \sum_{i=1}^{N} \|T_i \alpha_i \mathbf{c}_i - \overline{T}_i \overline{\alpha}_i \overline{\mathbf{c}}_i\|$$

$$\leq \sum_{i=1}^{N} |T_i - \overline{T}_i||\alpha_i|\|\mathbf{c}_i\| + |\overline{T}_i||\alpha_i - \overline{\alpha}_i|\|\mathbf{c}_i\| + |\overline{T}_i \overline{\alpha}_i|\|\mathbf{c}_i - \overline{\mathbf{c}_i}\|$$

$$\leq \sum_{i=1}^{N} (t_i \delta_i L + 2\delta_i L)\|\mathbf{o} - \overline{\mathbf{o}}\| + C$$

$$\leq 3(\sum_{i=1}^{N} \delta_i)L\|\mathbf{o} - \overline{\mathbf{o}}\| + C$$

$$= 3L\|\mathbf{o} - \overline{\mathbf{o}}\| + C.$$

The first inequality is by definition, the second inequality is by triangle inequality, the third inequality is by the conclusion we have proved and the bounding property of $T_i$, $\alpha_i$ and $\mathbf{c}_i$, the final inequality is by $t_i \leq 1$ and the final equality is by $\delta_i = t_{i+1} - t_i$ and $t_1 = 0, t_{N+1} = 1$. $C$ represents a bounding constant of $\|\mathbf{d} - \overline{\mathbf{d}}\|$, independent of $\|\mathbf{o} - \overline{\mathbf{o}}\|$. $\square$

# B EXPERIMENT DETAILS

## B.1 SPARSE VIEW SAMPLING

| PSNR↑ | hotdog | lego | chair | drums | ficus | materials | mic | ship | Avg. |
|---|---|---|---|---|---|---|---|---|---|
| NeRF + Rand | 22.19 | 19.85 | 19.99 | 10.93 | 18.13 | 8.73 | 17.85 | 15.31 | 16.62 |
| NeRF + FVS | 23.87 | 17.83 | 20.06 | 15.38 | 17.91 | 13.76 | 17.91 | 15.94 | 17.83 |
| ActiveNeRF | 17.87 | 18.96 | 20.20 | 14.82 | **22.55** | **18.19** | 17.92 | **19.34** | 18.73 |
| Ours (S→P) | **24.01** | 20.48 | **26.21** | 16.78 | 18.49 | 13.95 | 17.57 | 13.95 | 18.93 |
| Ours (P→S) | 23.14 | **22.90** | 20.08 | **17.96** | 20.99 | 15.16 | **24.01** | 16.50 | **20.09** |

| SSIM↑ | hotdog | lego | chair | drums | ficus | materials | mic | ship | Avg. |
|---|---|---|---|---|---|---|---|---|---|
| NeRF + Rand | 0.919 | 0.838 | 0.848 | 0.793 | 0.845 | 0.762 | 0.881 | 0.689 | 0.822 |
| NeRF + FVS | **0.922** | 0.798 | 0.853 | 0.776 | 0.838 | 0.776 | 0.879 | 0.706 | 0.819 |
| ActiveNeRF | 0.860 | 0.829 | 0.858 | 0.768 | **0.886** | **0.813** | 0.876 | 0.716 | 0.826 |
| Ours (S→P) | 0.918 | **0.852** | **0.898** | 0.793 | 0.848 | 0.789 | 0.883 | **0.789** | **0.846** |
| Ours (P→S) | 0.916 | 0.851 | 0.849 | **0.814** | 0.859 | 0.812 | **0.924** | 0.704 | 0.841 |

| LPIPS↓ | hotdog | lego | chair | drums | ficus | materials | mic | ship | Avg. |
|---|---|---|---|---|---|---|---|---|---|
| NeRF + Rand | 0.089 | 0.152 | 0.165 | 0.231 | 0.152 | 0.241 | 0.138 | 0.317 | 0.186 |
| NeRF + FVS | **0.082** | 0.197 | 0.158 | 0.239 | 0.167 | 0.205 | 0.140 | 0.304 | 0.186 |
| ActiveNeRF | 0.172 | 0.150 | 0.149 | 0.253 | **0.116** | **0.145** | 0.142 | 0.319 | 0.181 |
| Ours (S→P) | 0.089 | **0.135** | **0.109** | 0.218 | 0.152 | 0.177 | 0.139 | **0.177** | **0.149** |
| Ours (P→S) | 0.099 | 0.153 | 0.165 | **0.183** | 0.136 | 0.159 | **0.093** | 0.306 | 0.162 |

**Table 5: Quantitative comparison on Blender in Setting I.** We provide a detailed listing of the metric values for each object on Blender, which is the same in Table 1 in the manuscript.

We conduct experiments in Active Learning settings using the ActiveNeRF (Pan et al., 2022) codebase. In traditional NeRF (Mildenhall et al., 2020), we obtain a volume parameter $\sigma$ and color values $c = (r, g, b)$ for a specific position and direction. In ActiveNeRF, it simultaneously outputs both mean and variance, following a Gaussian distribution. For simplicity, we adopt the ActiveNeRF version and apply its pipeline to our baseline methods *(NeRF+Random, NeRF+FVS)* as well as our proposed strategy. The primary modification we make is in the evaluation step, which is central to this active learning setting.

Its original codebase only provides training configuration files for a portion of the LLFF dataset and the Blender dataset. We observe that for the Blender dataset, the codebase used a fixed number (20) of initial training samples so we cannot decide the initial training set size. We then modify it to allow the selection of the initial training set size, with the remaining images serving as a holdout set. For instance, in Setting I, for each object in the Blender dataset with 100 ordered images, we choose the first 4 images as the initial set and use the remaining 96 images as the holdout set. Due to excessive memory requirements, training on the LLFF dataset is not feasible even on a 48GB A40 GPU, so we temporarily refrain from conducting experiments on it. However, we believe that the results on the Blender dataset sufficiently validate our claims.

Due to the randomness of the strategy and potential variations in the training process, we conducted three experiments for each result and selected the average outcome. In Table 5, We provide a detailed breakdown of the specific results for each object on Blender in Setting I.

### B.2 FEW-SHOT VIEW SYNTHESIS

#### B.2.1 DATASET

We conduct our experiments in the few-shot setting across three datasets: the Blender dataset (Mildenhall et al., 2020), the DTU dataset (Jensen et al., 2014), and the LLFF dataset (Mildenhall et al., 2019). Many works focus on the few-shot setting using different benchmarks, making it challenging to compare all of them uniformly. To ensure a fair and comprehensive comparison, we adopt the settings from FreeNeRF (Yang et al., 2023). We conduct the experiments on a 48GB A40 GPU.

**Blender Dataset:** The Blender dataset (Mildenhall et al., 2020) comprises eight synthetic scenes. We follow the data split used in DietNeRF (Jain et al., 2021) to simulate a few-shot neural rendering scenario. For each scene, the training images with IDs (counting from "0") 26, 86, 2, 55, 75, 93, 16, 73, and 8 are used as the input views, and 25 images are sampled evenly from the testing images for evaluation.

**DTU Dataset:** The DTU dataset (Jensen et al., 2014) is a large-scale multiview dataset consisting of 124 different scenes. PixelNeRF (Yu et al., 2021b) uses a split of 88 training scenes and 15 test scenes to study the pre-training or per-scene fine-tuning setting in a few-shot neural rendering scenario. Unlike FreeNeRF, we do not require pre-training. We follow (Niemeyer et al., 2022) to optimize NeRF models directly on the 15 test scenes. The test scan IDs are 8, 21, 30, 31, 34, 38, 40, 41, 45, 55, 63, 82, 103, 110, and 114. In each scan, the images with the following IDs (counting from "0") are used as the input views: 25, 22, 28. The images with IDs in [1, 2, 9, 10, 11, 12, 14, 15, 23, 24, 26, 27, 29, 30, 31, 32, 33, 34, 35, 41, 42, 43, 45, 46, 47] serve as the novel views for evaluation. According to the FreeNeRF, masks of the DTU dataset do not always help improve PSNR and SSIM and sometimes the PSNR score in a specific scene drops a lot. For a fair comparison, we train one model for one scene to produce the results in the object and full-image setting at the same time.

**LLFF Dataset:** The LLFF dataset (Mildenhall et al., 2019) is a forward-facing dataset containing eight scenes. Adhering to (Mildenhall et al., 2020; Niemeyer et al., 2022), we use every 8th image as the novel views for evaluation and evenly sample the input views from the remaining views.

#### B.2.2 EXPERIMENT RESULTS

Figures 5 and 6 present qualitative results on the DTU and LLFF datasets, respectively, corresponding to the quantitative results in Table 3.

In our experiments, $L_{\text{micro}}$ represents the variance of the mean color value between training images and randomly rendered images, ensuring that the color difference is constrained within a certain range. This is based on Lemma 3, where we emphasize the color difference between images. $L_{\text{micro}}$ is not limited to this form and can be interpreted using other measures like KL-divergence in color, which can also achieve similar performance.

Similarly, $L_{\text{macro}}$ is not restricted to using CLIP. Other models such as DINO Caron et al. (2021) or BLIP Li et al. (2022) can also extract semantic features for our framework.

Our framework is flexible and can incorporate various forms of regularization terms related to semantic space distance or pixel space distance, allowing for broad applicability and adaptability.

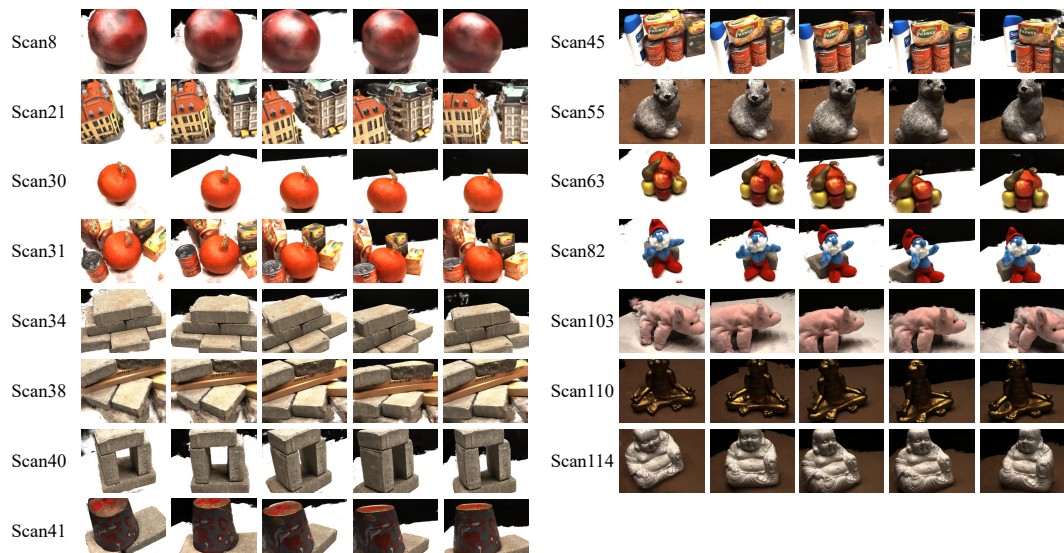

Figure 5: Example of our results with 3 input views on the DTU dataset.

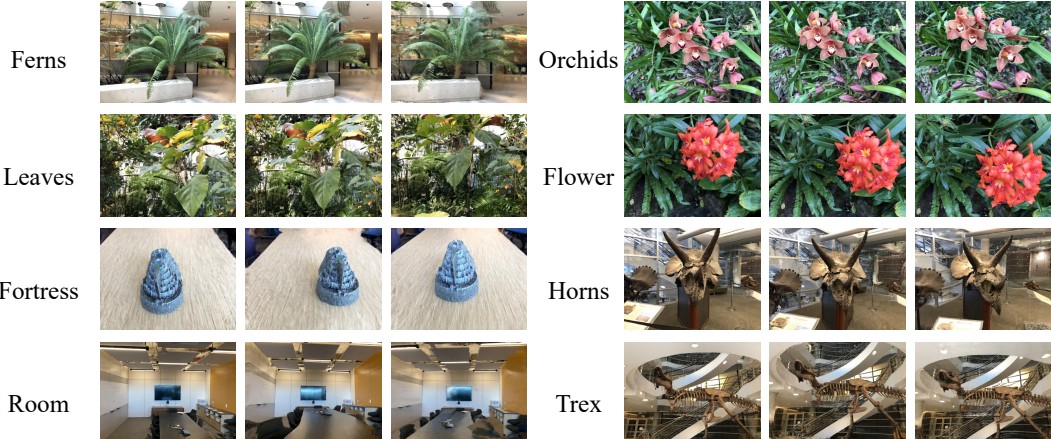

Figure 6: Example of our results with 3 input views on the LLFF dataset.

### B.2.3 LIMITATIONS ON BASELINES

FreeNeRF is a strong baseline that achieves state-of-the-art performance compared to methods using priors from diffusion models across many datasets. We get this conclusion from the experiment results of ReconFusion (Wu et al., 2023). Therefore, it is worthwhile to continue our comparison between our method and some diffusion-based methods like SparseFusion (Zhou & Tulsiani, 2022) or ReconFusion (Wu et al., 2023).

SparseFusion's evaluation is currently limited to the CO3D dataset (Reizenstein et al., 2021), and it lacks performance data on three popular and classical datasets which we have used to keep the same as FreeNeRF: the Blender dataset, the DTU dataset and the LLFF dataset. Fair evaluations of SparseFusion on these datasets are absent, and addressing this gap would require significant additional time, which might divert from our primary research focus. Nonetheless, the datasets we employ are robust and widely accepted in NeRF research, providing sufficient support for our experiments with numerous baseline performances available for reference.

Additionally, the lack of open-source code for ReconFusion limits our ability to apply custom regularization terms or conduct meaningful comparisons. Future work should aim to incorporate more new baseline methods and explore additional variations within our framework.

