# OpenReview forum: "MutualNeRF: Improve the Performance of NeRF under Limited Samples with Mutual Information Theory"
_ICLR.cc/2025/Conference — Submitted to ICLR 2025_

### Official Review · Reviewer_BHX7 · 2024-11-01

**Soundness:** 3
**Presentation:** 3
**Contribution:** 2
**Rating:** 5
**Confidence:** 3

**Summary:**

This paper proposes utilizing Mutual Information Theory to enhance the performance of NeRF with limited samples. Mutual information serves as a metric to measure the correlation between images at both macro and micro levels. The macro perspective focuses on correlations in semantic features, while the micro perspective addresses correlations in the pixel space. By incorporating mutual information, the proposed framework effectively addresses challenges posed by sparse view sampling and few-shot view synthesis, resulting in improved performance. Extensive experiments demonstrate the effectiveness of assessing both semantic and pixel space distances.

**Strengths:**

1. Introducing mutual information to enhance Neural Radiance Field (NeRF) performance is an innovative idea.
2. The experimental results in Table 3 show that the proposed method improves performance across various NeRF frameworks.
3. This paper is well-organized.

**Weaknesses:**

1. Results in Figure 4 show limited improvement between FreeNeRF and the proposed MutualNeRF; only the zoomed-in sections reveal marginal enhancements, which are still not substantial.
2. The proposed method introduces more complex training requirements and utilizes a large model, CLIP, for assessing semantic space distance, which may impact training time and peak memory consumption.
3. Performance improvements become marginal with more powerful baseline frameworks. For instance, "RegNeRF + Ours" surpasses RegNeRF by 1.28 PSNR, while "FreeNeRF + Ours" outperforms FreeNeRF by only 0.50 PSNR.

**Questions:**

1) Figure 4 shows that some regions, such as the stairs, are not improved. Why might this be the case? Could this indicate limitations in the Mutual Information design? Specifically, which types of regions can be enhanced by the macro and micro losses? Currently, it is unclear which specific issues the proposed framework addresses. Could the authors provide a more detailed explanation?

2) What are the comparisons in terms of training time and peak memory consumption?

3) Table 3 and Table 4 show that the performance gain for "FreeNeRF + Ours" is marginal. This raises concerns about whether MutualNeRF will improve performance for more advanced backbones, such as Sparsenerf [A] or Sparsenerf combined with FreeNeRF. Could the authors comment on this?

4) Does the proposed macro loss also improve the performance of 3DGS-based methods? If so, did the authors experiment with sparse-view 3DGS splatting methods, such as [B]?

5) What would happen if we applied CLIP as a perceptual loss to support FreeNeRF training?

---

> ### Author Response · Authors · 2024-11-19
> **Rebuttal by Authors**
>
> We extend our gratitude to the reviewer for the comments and suggestions. Below, we address the primary concerns that have been raised.
>
>
>
> >Q1: The proposed method introduces more complex training requirements and utilizes a large model, CLIP, for assessing semantic space distance, which may impact training time and peak memory consumption.
>
> **A1:** We have explored **alternative semantic metrics** to quantify "image similarity," such as DINOv2[1] and MAE[2], as encoders in our loss computation. Below are the comparative results with NeRF on the Blender dataset:
>
> | Method | PNSR ↑ | SSIM↑ | LPIPS ↓|
> | -------- | -------- | -------- | -------- |
> | NeRF     | 14.934    | 0.687    | 0.318 |
> | NeRF + CLIP (vit-base-patch32) | 22.503 (+7.569) | 0.823 (+0.136) | 0.124(-0.194) |
> | NeRF + DINOv2 (dinov2-base) | 22.882(+7.948)|0.830(+0.143) |0.119(-0.199) |
> | NeRF + MAE (MAE-ViT) | 21.652(+6.718) | 0.798(+0.111) | 0.165(-0.153)
>
>
> Our findings indicate that different encoders exhibit similar performance improvements, demonstrating that **our framework is generalizable** and not heavily rely on clip to calculate semantics.
>
> >Q2: Figure 4 shows that some regions, such as the stairs, are not improved. Why might this be the case? Could this indicate limitations in the Mutual Information design? Specifically, which types of regions can be enhanced by the macro and micro losses? Currently, it is unclear which specific issues the proposed framework addresses. Could the authors provide a more detailed explanation?
>
>
> **A2:** Thank you for pointing out the limitation shown in Figure 4. The observed lack of improvement in certain regions, such as the stairs, might be due to insufficient feature alignment between the synthetic and reference views in these areas. This issue may arise because such regions often contain intricate geometric or high-frequency texture details that are challenging for the mutual information-based design to fully capture, particularly when the macro and micro features have lower correspondence.
>
> Our proposed framework leverages macro and micro mutual information losses to enhance overall structure and fine-grained detail consistency, respectively. The macro loss is effective for improving large-scale structural features by maximizing high-level correspondence, whereas the micro loss helps to align local fine details, which might be missing or mismatched across views. We plan to investigate improved methods for enhancing intricate regions, such as adaptive feature extraction or incorporating geometry-aware constraints to better align these areas.
>
> >Q3: Does the proposed macro loss also improve the performance of 3DGS-based methods? If so, did the authors experiment with sparse-view 3DGS splatting methods?
>
> **A3:** Our paper primarily focuses on NeRF synthesis with limited training data. Due to the limited time for the rebuttal, we are unable to directly conduct experiments related to 3DGS. We believe extending this approach to 3DGS is a valuable direction for future work.
>
> >Q4: What would happen if we applied CLIP as a perceptual loss to support FreeNeRF training?
>
> **A4:** Applying CLIP as a perceptual loss to support FreeNeRF training could potentially improve the semantic consistency between the generated and real images, as CLIP's feature space is trained to align well with human perception. We think incorporating CLIP as a perceptual loss might introduce additional computational complexity, as observed with our integration of CLIP in the NeRF framework. In future experiments, we plan to further explore the application of CLIP to FreeNeRF to assess its effectiveness in improving the synthesis quality under sparse-view settings.
>
> We thank the reviewer once again for the valuable and helpful suggestions.
>
> **References**
>
> [1] Oquab M, Darcet T, Moutakanni T, et al. Dinov2: Learning robust visual features without supervision[J]. arXiv preprint arXiv:2304.07193, 2023.
>
> [2] He K, Chen X, Xie S, et al. Masked autoencoders are scalable vision learners[C]//Proceedings of the IEEE/CVF conference on computer vision and pattern recognition. 2022: 16000-16009.

---

> > ### Comment · Reviewer_BHX7 · 2024-11-26
> >
> > Dear Authors,
> >
> > Thank you for your detailed response. However, I noticed that some of my questions were not addressed directly. For instance, there is no comparison provided regarding training time or peak memory consumption. Additionally, the experiments in the rebuttal are not convincing, as the base method used is NeRF alone. I wonder why more advanced NeRF variants, such as FreeNeRF or RegNeRF, were not utilized for comparison.
> >
> > Based on these concerns, I tend to maintain my score.
> >
> > Best regards,

---

### Official Review · Reviewer_Hj8j · 2024-11-04

**Soundness:** 2
**Presentation:** 3
**Contribution:** 2
**Rating:** 5
**Confidence:** 4

**Summary:**

The paper introduces MutualNeRF, a framework that enhances Neural Radiance Field (NeRF) performance under limited sample conditions using Mutual Information Theory. It addresses sparse view sampling and few-shot view synthesis by minimizing and maximizing mutual information, respectively. The framework employs a greedy algorithm for viewpoint selection and plug-and-play regularization terms for training.

**Strengths:**

Utilizes mutual information theory to improve NeRF under limited data.

Proposes a greedy algorithm for strategic viewpoint selection in sparse view sampling.

Introduces efficient regularization terms to enhance few-shot view synthesis.

**Weaknesses:**

1.	How about using mutual information for 3DGS?
2.	What is the rendering speed of the proposed method, especially when compared with 3DGS?
3.	I suggest that the authors compare or discuss with more methods, like PixelNeRF[A], CR-NeRF[B], and MVSGaussian[C], to fully verify the effectiveness of the proposed mutual information.
4.	Why choose to pick images instead of training as a whole? Are there images that do not belong to the target scene? If we use a fast reconstruction method like 3DGS, training all images won't take much time.

**Reference**

[A] Yu, Alex, et al. "pixelnerf: Neural radiance fields from one or few images." Proceedings of the IEEE/CVF conference on computer vision and pattern recognition. 2021.

[B] Yang, Yifan, et al. "Cross-ray neural radiance fields for novel-view synthesis from unconstrained image collections." Proceedings of the IEEE/CVF International Conference on Computer Vision. 2023.

[C] Liu, Tianqi, et al. "MVSGaussian: Fast Generalizable Gaussian Splatting Reconstruction from Multi-View Stereo." European Conference on Computer Vision. Springer, Cham, 2025.

**Questions:**

Please refer to the weakness part.

---

> ### Author Response · Authors · 2024-11-19
> **Rebuttal by Authors**
>
> We thank the reviewer for the comments and constructive suggestions. In the following, we address the main concern raised. Please find the details below.
>
> >Q1: How about using mutual information for 3DGS?
>
> **A1:** Our paper primarily focuses on NeRF synthesis with limited training data. Due to the limited time for the rebuttal, we are unable to directly conduct experiments related to 3DGS. We believe extending the idea of mutual information to 3DGS is a valuable direction for future work.
>
>
> >Q2: What is the rendering speed of the proposed method, especially when compared with 3DGS?
>
>
> **A2:** We did not add this loss at every iteration. In fact, applying it every 100 iterations yields very good results, with the time overhead being less than 1.2 times.
>
>
>
> >Q3: Why choose to pick images instead of training as a whole? Are there images that do not belong to the target scene?
>
>
> **A3:** The issue of limited acquisition budget is **well addressed** in  ActiveNeRF[4] presented at ECCV 2022. The main idea of their paper is that NeRF usually requires a large number of posed images and generalize poorly with limited inputs. And it takes a whole observation in the scene to train a well-generalized NeRF. This poses challenges under real-world applications such as robot localization and mapping, where capturing training data can be costly, and perception of the entire scene is required.
>
> Our approach **strictly follows the acknowledged setup in the domains of active learning and NeRF**, ensuring a fair comparison with state-of-the-art (SOTA) methods.
>
>
>
> Finally, we thank the reviewer once again for the effort in providing us with valuable suggestions. We will continue to provide clarifications if the reviewer has any further questions.
>
>
> **References**
>
> [1] Yu, Alex, et al. "pixelnerf: Neural radiance fields from one or few images." Proceedings of the IEEE/CVF conference on computer vision and pattern recognition. 2021.
>
> [2] Yang, Yifan, et al. "Cross-ray neural radiance fields for novel-view synthesis from unconstrained image collections." Proceedings of the IEEE/CVF International Conference on Computer Vision. 2023.
>
> [3] Liu, Tianqi, et al. "MVSGaussian: Fast Generalizable Gaussian Splatting Reconstruction from Multi-View Stereo." European Conference on Computer Vision. Springer, Cham, 2025.
>
> [4] Pan X, Lai Z, Song S, et al. Activenerf: Learning where to see with uncertainty estimation[C]//European Conference on Computer Vision. Cham: Springer Nature Switzerland, 2022: 230-246.

---

> > ### Comment · Reviewer_Hj8j · 2024-12-03
> >
> > Thank you to the authors for their response. First, the data setup and training speed in the paper still lack a clear explanation. Second, a comparison with 3DGS is essential. Therefore, I will maintain my rating.

---

> ### Author Response · Authors · 2024-12-01
>
> Thanks again for your valuable feedback! Could you please let us know whether your concerns have been addressed? We are happy to make further updates if you have any other questions or suggestions.

---

### Official Review · Reviewer_ryB4 · 2024-11-04

**Soundness:** 3
**Presentation:** 3
**Contribution:** 2
**Rating:** 5
**Confidence:** 5

**Summary:**

This paper proposes a framework to improve the performance of sparse view sampling and few-shot view synthesis by mutual information. For sparse view sampling, the authors first calculate the semantic space distance and pixel space distance, then use a greedy algorithm to select sparse views for training. For few-shot view synthesis, the authors add two regularization terms based on semantic space distance and pixel space distance, to improve the performance of view synthesis. The authors conduct detailed experiments to demonstrate the effectiveness.

**Strengths:**

The authors demonstrate the effectiveness of both sparse view sampling and few-shot view synthesis. Using mutual information to select views for NeRF is reasonable.

**Weaknesses:**

1. The motivation for pixel space distance is not clarified. According to Definition 3, the pixel space distance is the expectation of distance between any two points of rays. The authors should clarify the motivation behind Definition 3 since it is important for the following parts. The semantic space distance is fairly reasonable.

2. If my understanding is correct, for the few-shot view synthesis, the authors just added two regularization terms to the NeRF training. However, the relationship between mutual information and few-shot view synthesis is not clear. For sparse view sampling, minimizing mutual information is reasonable.

3. For the few-shot view synthesis, the proposed method needs to evaluate the semantic distance between randomly rendered images and ground truth images. Therefore, this will bring additional training costs. The analysis should be provided.

4. Compared with NeRF, 3D Gaussian splatting is much better in both training and rendering. The authors should conduct experiments based on 3DGS.

5.  There are many typos and grammatical errors. For example, formulas should come with indices.

**Questions:**

please refer to the weaknesses part.

---

> ### Author Response · Authors · 2024-11-19
> **Rebuttal by Authors**
>
> We express our gratitude to the reviewer for the insightful comments. We address the main concerns below. If the reviewer believes there are additional issues with our paper that have not been addressed, we are very willing to continue the discussion.
>
> >Q1: The motivation for pixel space distance is not clarified. According to Definition 3, the pixel space distance is the expectation of distance between any two points of rays. The authors should clarify the motivation behind Definition 3 since it is important for the following parts. The semantic space distance is fairly reasonable.
>
> **A1:** We appreciate the reviewer's feedback. Our intuition is mainly based on the idea that capturing light rays from different positions of an object provides richer information about it. Since the object varies continuously internally, we measure the richness of the information provided by evaluating the differences at every point along two light rays.
>
> >Q2: The relationship between mutual information and few-shot view synthesis is not clear.
>
> **A2:** This paper primarily focuses on providing a new perspective for understanding sampling under limited data. Sparse view sampling and few-shot synthesis are just two application scenarios we explored. More application scenarios can be provided in future work.
>
>
> >Q3: For the few-shot view synthesis, the proposed method needs to evaluate the semantic distance between randomly rendered images and ground truth images. Therefore, this will bring additional training costs. The analysis should be provided.
>
> **A3:** We did not add this loss at every iteration. In fact, applying it every 100 iterations yields very good results, with the time overhead being less than 1.2 times.
>
> >Q4: Compared with NeRF, 3D Gaussian splatting is much better in both training and rendering. The authors should conduct experiments based on 3DGS.
>
> **A4:** Our paper primarily focuses on NeRF synthesis with limited training data. Due to the limited time for the rebuttal, we are unable to directly conduct experiments related to 3DGS. We believe extending the idea of mutual information to 3DGS is a valuable direction for future work.
>
>
>
> We would be delighted to address any further inquiries you may have. Please feel free to reach out with any additional questions or concerns!

---

> > ### Comment · Reviewer_ryB4 · 2024-11-25
> > **Re: Rebuttal by Authors**
> >
> > The reviewer expresses gratitude to the authors for their rebuttal.
> >
> > However, I am still a bit confused about the motivation behind "the richness of the information provided by evaluating the differences at every point along two light rays". I recommend that the authors present a clearer explanation. Besides, The current lack of experiments related to 3DGS makes the proposed approach less significant at this time.
> >
> > Therefore, I will maintain my rating.

---

### Official Review · Reviewer_MJVb · 2024-11-04

**Soundness:** 2
**Presentation:** 2
**Contribution:** 2
**Rating:** 3
**Confidence:** 4

**Summary:**

The paper proposes MutualNeRF, a method to improve the performance of Neural Radiance Fields (NeRF) when training samples are limited. The authors integrate mutual information theory to develop a unified approach, enhancing NeRF's effectiveness in sparse view sampling and few-shot view synthesis, by modeling uncertainty in semantic space distance and pixel space distance.

**Strengths:**

- This paper is well written and easy to follow.
- The framework’s design is comprehensive, considering both macro (semantic) and micro (pixel) perspective in the task of sparse view sampling and few-shot NVS.

**Weaknesses:**

- **Complex methodology with marginal gains:** This methodology introduces significant complexity, especially in sparse view sampling, involving greedy algorithms and complex mutual information metrics. However, the observed improvements over simpler baselines are relatively minor, which may not justify the added complexity.
- **Lack of novelty.** The attempt to address the task of few-shot novel view synthesis through minimization of mutual information between viewpoints have already been explored, especially in CVRP 2022 paper **InfoNeRF: Ray Entropy Minimization for Few-Shot Neural Volume Rendering**. The methodology of this paper seems to be re-interpretation of ideas used in DietNeRF (semantic space) and InfoNeRF (pixel space) within the perspective of mutual information theory. I ask the authors to give a more thorough theoretical comparison between this work and the methods that I have mentioned.

**Questions:**

See the weaknesses above.

---

> ### Author Response · Authors · 2024-11-19
> **Rebuttal by Authors**
>
> We greatly appreciate the reviewer's comments and valuable suggestions.
>
> >Q1: This methodology introduces significant complexity, especially in sparse view sampling, involving greedy algorithms and complex mutual information metrics. However, the observed improvements over simpler baselines are relatively minor, which may not justify the added complexity.
>
> **A1:** We appreciate the reviewer's feedback. The primary goal of this paper is to provide a novel perspective on the problem of sampling under limited data. We identified specific application scenarios within two NeRF setups and demonstrated performance improvements over baseline algorithms in both cases. Additionally, more application scenarios can be explored in future work.
> We followed the settings of ActiveNeRF. Since we only need to compute CLIP similarity and camera distance, the time overhead is only 1/5 of that of ActiveNeRF.
>
>
> >Q2: The attempt to address the task of few-shot novel view synthesis through minimization of mutual information between viewpoints have already been explored, especially in CVRP 2022 paper InfoNeRF: Ray Entropy Minimization for Few-Shot Neural Volume Rendering. The methodology of this paper seems to be re-interpretation of ideas used in DietNeRF (semantic space) and InfoNeRF (pixel space) within the perspective of mutual information theory. I ask the authors to give a more thorough theoretical comparison between this work and the methods that I have mentioned.
>
> **A2:** We thank the reviewer's comment. First, we provide two application scenarios: sparse view sampling and few-shot synthesis. For the second scenario, our approach is entirely different from that in the InfoNeRF[1]. InfoNeRF uses Shannon Entropy to define the entropy of a discrete ray density function, while we base our method on concepts from mutual information, which involve significant conceptual differences.
>
>
> **References**
> [1] Kim, Mijeong, Seonguk Seo, and Bohyung Han. "Infonerf: Ray entropy minimization for few-shot neural volume rendering." Proceedings of the IEEE/CVF Conference on Computer Vision and Pattern Recognition. 2022.

---

> > ### Author Response · Authors · 2024-12-01
> >
> > Thanks again for your valuable feedback! Could you please let us know whether your concerns have been addressed? We are happy to make further updates if you have any other questions or suggestions.

---

### Author Response · Authors · 2024-11-23

We would like to express our sincere gratitude for the reviewer's constructive suggestions and comments. Since the deadline is approaching, we sincerely hope the reviewers can read our response. Please let us know if the reviewers have any comments about our response or any other additional concerns. We are eager to provide any further clarifications and discussions to help the evaluation.

---

### Meta-Review · Area_Chair_nrfe · 2024-12-22

**Metareview:**

This paper received negative feedback from the reviewers, who expressed concerns regarding the lack of novelty and the limited comparisons. Specifically, the use of mutual information for few-shot novel view synthesis has been explored in previous methods. The AC has carefully reviewed all the feedback and the authors' rebuttal and agrees that the current version is not yet ready for acceptance. The AC strongly encourages the authors to address all the reviewers' concerns and resubmit to a future venue.

**Additional Comments On Reviewer Discussion:**

There were no reviewer discussions.

---

### Decision · Program_Chairs · 2025-01-22

Reject